# Optimal Selection of Sampling Points within Sewer Networks for Wastewater-Based Epidemiology Applications

**DOI:** 10.3390/mps7010006

**Published:** 2024-01-05

**Authors:** Yao Yao, Yibo Zhu, Regina Nogueira, Frank Klawonn, Markus Wallner

**Affiliations:** 1Institute for Information Engineering, Ostfalia University of Applied Sciences, Salzdahlumer Str. 46/48, 38302 Wolfenbüttel, Germany; yao.yao1@ostfalia.de; 2Faculty of Civil and Environmental Engineering, Ostfalia University of Applied Sciences, Herbert-Meyer-Str. 7, 29556 Suderburg, Germany; yibo_zhu@web.de (Y.Z.); m.wallner@ostfalia.de (M.W.); 3Institute of Sanitary Engineering and Waste Management, Leibniz University Hannover, Welfengarten 1, 30167 Hannover, Germany; nogueira@isah.uni-hannover.de; 4Biostatistics Research Group, Helmholtz Centre for Infection Research, 38124 Braunschweig, Germany

**Keywords:** optimal sampling point, wastewater-based epidemiology, information theory, pathogen surveillance

## Abstract

Wastewater-based epidemiology (WBE) has great potential to monitor community public health, especially during pandemics. However, it faces substantial hurdles in pathogen surveillance through WBE, encompassing data representativeness, spatiotemporal variability, population estimates, pathogen decay, and environmental factors. This paper aims to enhance the reliability of WBE data, especially for early outbreak detection and improved sampling strategies within sewer networks. The tool implemented in this paper combines a monitoring model and an optimization model to facilitate the optimal selection of sampling points within sewer networks. The monitoring model utilizes parameters such as feces density and average water consumption to define the detectability of the virus that needs to be monitored. This allows for standardization and simplicity in the process of moving from the analysis of wastewater samples to the identification of infection in the source area. The entropy-based model can select optimal sampling points in a sewer network to obtain the most specific information at a minimum cost. The practicality of our tool is validated using data from Hildesheim, Germany, employing SARS-CoV-2 as a pilot pathogen. It is important to note that the tool’s versatility empowers its extension to monitor other pathogens in the future.

## 1. Introduction

Wastewater-based epidemiology (WBE) provides near real-time information on public health status at the community level concerning specific pathogens and can potentially be a powerful tool for fighting pandemics [1,2,3]. However, studying pathogens based on WBE encompasses several significant challenges related to data representativeness, spatial and temporal variability, accurate population estimation, pathogen decay and dilution, and environmental confounders, strongly influencing virus detection capability. System sensitivity is introduced in this paper based on the mass balance model to standardize the monitoring process, which determines the detectability of sampling points in sewer systems. Utilizing reliable information from wastewater can help us harness the full potential of WBE to better comprehend and treat public health issues.

Due to the high cost and difficulties, only the inflow of wastewater treatment plants (WWTP), the endpoint of a sewer network, is generally used for WBE. However, this has various drawbacks. Signals from different catchments with different wastewater matrices are mixed beside the highly diluted inflow. Environmental conditions such as wastewater composition, temperature, or pH might impact virus decay and detection [4,5]. According to [6,7,8,9], there is an apparent information gain in sampling within the sewer network, rather than only at the endpoints. Thus, this paper aims to use sampling points in the sewer networks besides the inflow of WWTP. The sewer network is a system of pipes and manholes. Generally, the network is distinguished in separate sewer systems, where only wastewater from the population and industry is collected, and combined sewer systems, where additional rainfall is drained within the same pipes.

We develop an entropy-based model to select optimal sampling points to obtain the most specific information using as few sampling points as possible to generate information about pathogens within a community, such as virus distribution, concentration, and developing trends. SARS-CoV-2, as a typical large epidemic, still affects the world today. Our approach is applied to it as a pilot parameter. The general procedure is meant to apply to any other pathogen to facilitate the early detection of outbreaks and the optimization of sampling strategies for profound success in public health interventions.

The structure of our tool combines a monitoring model to detect positive signals from regions with infected populations and an optimization program to select optimal sampling points in the sewer system. Previous studies have focused on the real-time identification of patient zero. Ref. [10] developed two theoretical methods based on binary search algorithms to identify hotspots and patient zero in real-time. However, the presented approach depends strongly on rapid wastewater testing at each target sampling point. Up to now, no rapid wastewater testing is available. Some studies have addressed the general problem of identifying optimal sampling points. In [11], two algorithms based on graph theory combined with greedy optimization were proposed to select sampling points based on approaches by [12]. However, the problem of dilution effects and other parameters, which might impact the detectability of SARS-CoV-2, is only marginally addressed in the studies mentioned above. In [13], a tool that transforms the problem into a min–max problem based on allocating population to a sewer network was designed. The sampling points for this network are minimized by maximizing the covering discharges. However, dilution effects still influence this tool, and its efficiency is only guaranteed in small cities. In [14], the initial concentration and decay rate of SARS-CoV-2 on the detection time and detection likelihood of the virus at downstream nodes were explored. Tools that can identify optimal sampling points were also developed. However, the results and tool only remain applicable to cities of less than 50,000 people. Our approach, in combination with the network topology and the settlement structure, can select the optimal number of sampling points according to the system sensitivity. The system sensitivity is defined as standardizing the virus detectability in the wastewater data, i.e., the minimal number of infected people needed to detect a positive signal. The system’s applicability was tested in Hildesheim, Germany, with approximately 104,000 inhabitants.

This paper is structured as follows. Section 2 introduces the study area and data utilized in our research. Moreover, Section 2 outlines the general procedure and the specific parameters related to SARS-CoV-2, along with their uncertainties. It also introduces a mathematical approach grounded in information theory, which establishes connections between settlement structure as represented by residents and sewer topology. Section 3 presents the results, offering a practical perspective on the findings. Finally, in Section 4 we delve into the main discoveries of this study and explore potential avenues for future research.

## 2. Materials and Methods

### 2.1. Study Area

Hildesheim is a large city in northern Germany with approximately 104,000 residents. The urban catchment of Hildesheim is divided into 47 sub-catchments in this study based on the sewer network topology (Figure 1). Its main sewer network, which connects the sub-catchments via potential sampling points called candidate nodes, contains approximately 50 km of pipes. Wastewater flows through the sewer network to the northern part of the catchment. Then, it enters the Innerste River, receiving the catchment’s water after the WWTP. As the receiving water is not part of our system, it is not discussed further here.

The city has two sewer networks: (i) a combined system with combined sewer (~135 km) in the center part of the city and (ii) a separate system with wastewater sewer (~270 km) and rainwater sewer (~300 km) in the outer districts.

### 2.2. Data

#### 2.2.1. Sampling Data

First, for composite samples, we chose automatic samplers for sample collection. However, automatic samplers are expensive to acquire, maintain, and install, and sampling anywhere in the sewer system is labor- and equipment-intensive (see Figure 2a,b). Therefore, it is critical to select the ideal sampling point possible.

Second, we need selected manholes to be evaluated for overflow before installing the autosampler, as overflow could damage the autosampler. Nevertheless, there is still a risk (Figure 2c,d). Therefore, potential overflow can be used as a further selection criterion in the future, as long as the autosampler is used.

Finally, we also looked at comparing and selecting suitable samples. This is because the quality of the samples (e.g., some sites are prone to toilet paper clogging) and the different wastewater matrices (Figure 2e) can affect the detection of RNA. For calculating the incidences with the RNA concentrations in the water samples, normalization approaches, e.g., by using COD or biomarkers such as CrAssphage, exist to take the impact of the wastewater matric into account. In some cases, e.g., heavy rainfall events, a sampling is not recommended.

#### 2.2.2. Geographic Data

In this study, the population for each sub-catchment is derived based on the method of estimation of local population density in urban areas introduced by [15], i.e., the number of residents was proportional to the size of living space (the multiplication of building area at the residential area and building height) as the detailed population distribution is only available at the district level (14 districts in total).

Some sub-catchments in the northern part consist mainly of industry. This study does not include these sub-catchments as we focus primarily on domestic wastewater. In total, 63 candidate nodes are defined according to the sewer network’s topology. They can be divided into two groups: (i) outlet of sub-catchments (named with capital letter “S” and number) and (ii) intersection nodes in the main sewer (named with capital letters other than “S” and number). Table 1 shows the data used for this study and their sources.

### 2.3. General Procedure

The general procedure applied to optimize the needed number and location of sampling points in sewer networks for WBE contains two sequential steps. In the first step, the system sensitivity, i.e., the detectability of the SARS-CoV-2 virus, is defined. The sensitivity refers to a mass balance model utilizing equations from [16], covering the entire process, from virus RNA shedding through transport in the sewer system to wastewater sampling analysis in the lab. The mass balance model facilitates the sewer processes, providing sufficient results for further development. In the second step, an entropy-based mathematical model is formulated to optimize the selection of sampling points (manholes) in the sewer network for WBE applications.

### 2.4. System Sensitivity

The theoretically minimal number of infected individuals to detect positive signals in a region defines the system sensitivity, calculated based on the mass balance model. The model is presented in Figure 3, including equations from virus shedding to transport in the sewer network and sample analysis, considering the limit of detection of RNA fragments. In this study, RNA concentration in wastewater is calculated only on RNA in stool [17].

Merging all variables from the mass balance model and rearranging allows us to calculate $E_{inf}^{MIN}$ to mark a positive signal in a wastewater sample:(1)$$E_{inf}^{MIN}=\frac{c_{crit}*Q_{DWF}}{p_{s}*q_{s}*M_{s}*e^{-kt}}.$$

The stool volume, $q_{s}$, is calculated based on the feces production rate, feces density, and average water consumption (see details in Table 2). The dry weather flow, $Q_{DWF}$, only considers domestic sewage, assuming the sewers are in good condition without sewer infiltration and exfiltration. Moreover, the samples were taken on Sundays, with only a few industrial activities and impacts on wastewater runoff.

Some other variables are virus-specific, namely $c_{crit}$, $p_{s}$, $M_{s}$, and k, and their values were obtained via a literature review (Table 3). A summary of their values in different literature references can be found in Appendix A Table A1.

### 2.5. Optimization of Sampling Point Location

Information theory can evaluate the degree of dependence or redundancy between monitors [21] and is widely applied to the design and/or evaluation of monitoring networks for hydrological applications [22]. This section will elaborate on its mathematical background for selecting the optimal location of sampling points in sewer networks.

#### 2.5.1. Information Theory

Information is always viewed as a reduction in uncertainty. The Shannon entropy developed in information theory serves as a measure of information [23]. The information of event c with probability p(c) is expressed as $-\log_{2}p(c)$ [24].

In this study, the entropy is understood as the information capacity of signals and can be calculated for any random variable with a finite domain [25]. Different entropies and entropy-related measures are used to quantify the information content, namely entropy, joint entropy, and total correlation, as shown in Figure 4. Suppose circles A and B in Figure 4 are two sampling points. The size of each circle represents the gained information content.

Figure 4b illustrates the entropy for each sampling point, $C_{i}$. In this example, the entropy of B is larger than A’s, which means that B provides more information (reduces the uncertainty more substantially) than A. Formally, the entropy, $H\left(C_{i}\right)$, of a random variable, $C_{i}$, can be calculated by
(2)$$\begin{array}{c} H\left(C_{i}\right)=-\sum_{j=1}^{n_{i}}p\left(c_{i}^{j}\right)\log_{2}p\left(c_{i}^{j}\right),\;\;\;\;\sum_{i=1}^{n_{i}}p\left(c_{i}^{j}\right)=1 \end{array},i\in \left\{1,N\right\},$$
where N is the number of random variables (in our case, the number of sampling points) and $n_{i}$ is the number of all expected elementary events of random variable $C_{i}$ with values $c_{i}^{j}$ and their related probability distribution, $p\left(c_{i}^{j}\right)$. According to [24], the base of 2 in the logarithm is justified by the expected answers considering the monitoring location design, which is either “select” or “do not select” a sampling point.

Figure 4c represents the joint entropy, which shows the information content covered by both sampling points. If *N* random variables (*C*_1_, *C*_2_, …, *C_N_*) are considered, the total information content can be calculated by the joint entropy $H\left(C_{1},\;C_{2},\;\ldots ,\;C_{N}\right)$, which is defined as
(3)$$\begin{array}{c} H\left(C_{1},\;C_{2},\;\ldots ,\;C_{N}\right)=-\sum_{j_{1}=1}^{n_{1}}\sum_{j_{2}=1}^{n_{2}}\ldots \;\sum_{j_{N}=1}^{n_{N}}\;p\left(c_{1}^{j_{1}},\;c_{2}^{j_{2}},\;\ldots ,\;c_{N}^{j_{N}}\right)\log_{2}p\left(c_{1}^{j_{1}},\;c_{2}^{j_{2}},\;\ldots ,\;c_{N}^{j_{N}}\right) \end{array},$$
where $c_{i}^{j_{i}}$ is the $j_{i}$-th elementary event of random variable $C_{i}$, ($n_{1},\;n_{2},\;\ldots ,\;n_{N})$ are the numbers of elementary events of corresponding variables (*C*_1_, *C*_2_, …, *C_N_*), and $p\left(c_{1}^{j_{1}},\;c_{2}^{j_{2}},\;\ldots ,\;c_{N}^{j_{N}}\right)$ is the joint probability of events $\left(c_{1}^{j_{1}},\;c_{2}^{j_{2}},\;\ldots ,\;c_{N}^{j_{N}}\right)$.

Another interesting measure in information theory is the total correlation $TC\left(C_{1},\;C_{2},\;\ldots ,\;C_{N}\right)$, which describes the shared information amount of N random variables. The total correlation can be expressed by the difference between the individual entropies and the joint entropy,
(4)$$\begin{array}{c} TC\left(C_{1},\;C_{2},\;\ldots ,\;C_{N}\right)=\sum_{i=1}^{N}H\left(C_{i}\right)-H\left(C_{1},\;C_{2},\;\ldots ,\;C_{N}\right) \end{array},$$
where $\sum_{i=1}^{N}H\left(C_{i}\right)$ describes the sum of entropy of *N* random variables, ($C_{1},\;C_{2},\;\ldots ,\;C_{N}$), and $H\left(C_{1},\;C_{2},\;\ldots ,\;C_{N}\right)$ stands for the joint entropy, calculated by Equation (3). Figure 4d illustrates the redundant information (total correlation) of two sampling points (the overlapped area of two circles).

#### 2.5.2. Probability Distribution

As mentioned in the previous section, the key factor in information theory is the underlying probability distribution, which must be determined for individual problem domains. In most previous research works [27,28,29,30], it was derived from analyzing information through hydrodynamic simulations of the sewer network, including mass transport. In this study, a pragmatic approach combining information from simulations with network topology and settlement structure is developed to define the probability distribution. Figure 5 illustrates the probability distribution of two sub-catchments. For simplicity, it depends only on the number of residents in each sub-catchment. Nonetheless, additional factors from the epidemiological point of view, such as demography, socioeconomics, and population density, can be easily included in formulating the probability distribution on demand.

The tools developed are intended to control outbreaks at an early stage or even prevent them altogether through early detection. In the simplest situation, we search for “patient zero”. The probability distribution is simply a step function (Figure 5a). The probability, $p(X_{i})$, that the positive (infected) signal comes from the sub-catchment, $X_{i}$, is expressed by
(5)$$p(X_{i})=\frac{N_{i}}{\sum_{i=1}^{n_{x}}N_{i}},$$
where $N_{i}$ is the population of the sub-catchment, $X_{i}$, and $n_{x}$ is the total number of sub-catchments in the city. The infected probability for each sub-catchment is calculated based on the binomial theorem. However, due to the dilution effect, decay of the virus RNA in wastewater, and the detection limit of the analytical method, etc., the virus RNA load from one single patient may not be detected at the sampling point. Therefore, the input must be seen as a hotspot with more individuals infected. In this situation, Figure 5b shows a more rational definition of hotspots via probability distributions across sub-catchments. In this study, we assume that all potentially infected individuals belong to one sub-catchment, so a hotpot can be regarded as “patient zero”. The influences of the dilution effect, etc., are quantified by Equation (1) to define the detectability of each candidate node. Thus, Equation (5) can be used in this study, which will be justified with the results later.

#### 2.5.3. Signal Matrix and Entropy

As mentioned previously, system sensitivity represents the influence factors of the virus RNA load, such as the dilution effect, virus RNA decay in wastewater, and the detection limit of the analytical method. The RNA load determines whether a sampling point can detect the infected (positive) signal. At a specific system sensitivity, the potential signals detected by each sampling point form a signal matrix. Moreover, the signal matrix combines the information from the network topology and the settlement structure. We use a hypothetical sewer network with 2500 residents and six sub-catchments, as shown in Figure 6, to explain our approach. For simplicity, only main sewers are considered in this study.

The candidate nodes S1 to S6 for the sampling are pre-selected according to the sewer network topology, including the outlets of sub-catchments X1 to X6. Further candidate nodes A to E exist at the main sewer’s intersections.

As mentioned in Section 2.5.2, we assume all patients belong to only one sub-catchment because we aim to prevent or detect an outbreak early. The system sensitivity determines whether a positive signal can be detected in the candidate nodes for this sub-catchment, not only based on network topology. Besides potential signals, the probability of one sub-catchment being infected is shown in Table 4. The probability is calculated by residential density (See Equation (5)). For example, the “infected individual” belonging to sub-catchment X3 occurs with a probability of 0.26 and leads to a signal from candidate nodes S3 (the sampling point of this sub-catchment), D, and B (according to network topology) at this specific system sensitivity, 1:2200. Although candidate node A must theoretically show a positive signal based on network topology, considering other influence parameters (such as dilution effect) quantified by system sensitivity, it does not show a signal. A signal matrix can be built (see Table 4) to represent the potential signals (positive) of each sub-catchment when it is the source of the outbreak and no signals (negative) when not detected to be infected.

After constructing the signal matrix, the entropy calculation is formulated to find optimal sampling points covering the highest information content. For this purpose, we quantify their importance according to Equation (1),
(6)$$H\left(C_{i}\right)=-\sum_{j=1}^{n_{i}}p\left(c_{i^{+}}^{j}\right)\log_{2}p\left(c_{i^{+}}^{j}\right)-\left(1-\sum_{j=1}^{n_{i}}p\left(c_{i^{+}}^{j}\right)\right)\log_{2}\left(1-\sum_{j=1}^{n_{i}}p\left(c_{i^{+}}^{j}\right)\right),$$
where $p\left(c_{i^{+}}^{j}\right)$ is the probability of candidate node $C_{i}$ showing a potential positive signal because of an infected sub-catchment $X_{j}$, $n_{i}$ is the total number of sub-catchments (same by all candidate node $C_{i}$ in this study), and $\left(1-\sum_{j=1}^{n_{i}}p\left(c_{i^{+}}^{j}\right)\right)$ is the probability of this candidate node $C_{i}$ detecting no signals. Note that if no signal is detected in all sub-catchments, the probabilities of all combinations without a signal are summarized for the entropy calculation.

#### 2.5.4. Objective Function and Optimization Algorithm

An objective function called maximum information and minimum redundancy (MIMR) as an entropy measure for the optimization process is used by several studies [21,27,29]. In this study, a simplified form of the MIMR criterion based on [30] is applied,
(7)$$\begin{array}{c} {MIMR}_{C}= \end{array}\lambda_{1}H\left(C_{1},\;C_{2},\;\ldots ,\;C_{N}\right)-\lambda_{2}TC\left(C_{1},\;C_{2},\;\ldots ,\;C_{N}\right)\to MAX,where\;C=\left\{C_{1},\;\ldots ,\;C_{N}\right\},$$
where $H\left(C_{1},\;C_{2},\;\ldots ,\;C_{N}\right)$ is the joint entropy of N sampling points (see Equation (3)), $TC\left(C_{1},\;C_{2},\;\ldots ,\;C_{N}\right)$ is the total correlation (see Equation (4)), and $\lambda_{1}$ and $\lambda_{2}$ are the information redundancy weights to provide weights for joint entropy and total correlation ($\lambda_{1}+\lambda_{2}=1$).
(8)$$\begin{array}{c} \arg\underset{S\subseteq C}{\max}{MIMR}_{S} \end{array},\;\left|S\right|\;\leq z,$$
where z is the pre-defined maximum number of sampling points and S is the set of selected sampling points.

The optimization of Equation (7) leads to a monitoring network with a minimal number of sampling points (set S) from original sampling points (set C), where the information content of these sampling points has been maximized, while redundant information among these sampling points is minimized. To find the optimal solution to Equation (7), the MIMR-based greedy selection algorithm derived from [30] is adapted (see the pseudo codes Algorithm 1) to select sampling points iteratively (see Equation (8)). The first sampling point, S1 , is chosen based only on the entropy, HCi, where no total correlation is considered. The MIMR criterion is applied from selecting the second sampling point until the theoretical maximum joint entropy (JEMAX) is reached, or a defined number of sampling points (z) is achieved.
**Algorithm 1.** MIMR-based greedy selection algorithm to find optimal sampling points.**1****Procedure** FIND_OPTIMAL_S (C, N, X, S, z)**Input**: candidate set C including all N candidate nodes, catchment set X including all sub-catchments, and desired number of candidate nodes z.**Output**: selection set S including optimally selected candidate nodes.2Initialize maximum joint entropy JEMAX←HX using Equation (3), selection set S←∅, temporary joint entropy JETEMP←NULL.3**for** c∈C **do**4      Calculate entropy Hc for each candidate node using Equation (6)5**end for**6Assign optimal candidate node s←arg⁡maxc⁡[Hc]
7Update C←C\{s}
8Update S←S⋃{s}
9**while** ${JE}^{TEMP}\neq {JE}^{MAX}$ and $\left|S\right|\leq z$ **do**10      **for** c∈C **do**11            Calculate ${MIMR}_{S⋃\left\{c\right\}}$ using Equation (7)12      **end for**13            Find local optimal candidate node $s\leftarrow \arg\underset{c}{\max}{[MIMR}_{S⋃\left\{c\right\}}]$ using Equation (8)14            Update C←C\{s}
15            Update S←S⋃{s}
16            Assign temporary joint entropy ${JE}^{TEMP}\leftarrow H\left(S\right)$
17**end while**18**return** selection set S

## 3. Experimental Results and Discussion

We determine the minimum number of infected individuals required to trigger a positive signal at specific candidate nodes within the sewer network in Section 3.1. This forms the foundational premise for the subsequent discussion of the optimization results of sampling points within the catchment in Hildesheim in Section 3.2.

### 3.1. Determination of System Sensibility

The theoretical minimal number of infected people for each candidate node to obtain a positive signal, $E_{inf}^{MIN}$, must be determined before the optimization of sampling points by actual infectious number, which can be calculated using Equation (1). However, the used values of virus-dependent parameters for this equation vary between studies, as mentioned in Table 3. Thus, different combinations of virus-dependent parameter values and their impact on detecting SARS-CoV-2 RNA in wastewater were analyzed in this experiment based on a longitudinal section of the sewer network with nine candidate nodes, as shown in Figure 7. An average flow velocity of 0.5 m/s and the longest flow path from the main sewer was used to calculate the transport time.

The impact of the variability of the different virus-dependent parameter values was analyzed by two methods. The first was changing only one parameter between its min and max values by holding all other parameters with their median values from Table 3. The resulting minimal required number of infections are shown in Figure 7a. The decay value (*k*) influenced the results marginally, as the virus typically has only a short residential time until the sample is taken. The parameter with the most significant uncertainty is the shedding magnitude (*M_S_*), as values from the literature vary exponentially. Focusing on the *M_S_* in Figure 7a, at candidate node “A” the number of infected individuals needed to detect a positive signal exceeded the total population. This means no positive signal would be detected even if the entire population was infected. Compared to the median values, their optimal values from Table 3 were utilized in the second method. The results are illustrated in Figure 7b. With the optimal parameter combination, circa 40 infected individuals are needed in the catchment (104,231 residents) to detect a positive signal. In other words, one infected individual out of 2641 noninfected individuals could be detected under the combination of optimal parameter values. This value is close to our experience with real-world data, where SARS-CoV-2 RNA was detected in wastewater samples from the WWTP for incidences of approximately 40 infected individuals per 100,000 residents (1 infected individual in 2500). Thus, the optimal parameter values were used for the following analysis in Section 3.2.

Regardless of the virus-dependent parameter values, it can be observed in Figure 7 that more infected people are required to detect a positive signal in the downstream nodes of the sewer network compared to the upstream nodes. This can be attributed to the dilution effect. The downstream nodes are connected to more sub-catchments and residents, so more domestic water will be produced.

### 3.2. Optimization of Sampling Points

For the optimization of the MIMR (Equation (8)), the underlying optimal set of virus-specific parameters determined by the previous section is highlighted in Table 3. Furthermore, regarding the information redundancy weights, we gave more weight to the joint entropy than to the total correlation. Thus, $\lambda_{1}$ is chosen to be 0.8 and $\lambda_{2}$ as 0.2.

As introduced in Section 2.5.3, the signal matrix demonstrates the detected signals to calculate entropy for selecting optimal sampling points, which depends on the number of infected individuals in one sub-catchment and the $E_{inf}^{MIN}$. The system sensitivity in this study is 1:2641, as determined in Section 3.1. Table 5 shows the optimal number of sampling points depending on the number of infected individuals. The first column indicates the number of infected individuals. The second column depicts the number of minimal selected candidate nodes. The third and fourth columns show the maximum joint entropy and its relation to the theoretical maximum joint entropy of the covered area. The fifth and sixth columns demonstrate the covered population with the sampling points and the percentage of covered people in the entire population of the catchment.

According to Equation (1), the bigger the $E_{inf}$, the higher the load and the concentration of RNA. Therefore, reducing $N_{S}$ generally comes with a larger $E_{inf}$. The values in brackets in the column $N_{covered}$ resulted from probability distribution and entropy.

The entropy for each candidate node and different numbers of infected individuals ($E_{inf}=3,10,24,40$) are shown in Figure 8 to demonstrate the impact of the regional distribution of candidate nodes. For low RNA concentrations (small $E_{inf}$), the highest entropy is reached close to the outlets of the sub-catchments as further downstream dilution effects would reduce the detectability of SARS-CoV-2 in the wastewater. Therefore, if an early warning system is built to detect a small number of infected individuals within a catchment, more sampling points close to the sub-catchment outlets would be needed. Increasing RNA concentration (higher $E_{inf}$) allows fewer sampling points to cover more citizens and areas. For the case of 40 infected individuals, the problem of redundant information can be seen for the left main sewer close to the outlet. While the entropy indicated a high information content of the nodes at the end of this main sewer, it was evident that the information gain would only be minimal when adding two sampling points in a row (see red circles in Figure 8). This underlines the importance of including the total correlation as an additional measure in the objective function ${MIMR}_{C}$ (see Equation (7)).

The case of ten infected individuals and eight selected optimal candidate nodes is discussed in more detail. It was seen that the ${JE}^{MAX}$ is reached, and the whole population could be observed for this combination (see bold values in Table 5). The selected candidate nodes with corresponding sub-catchments are shown in Figure 9.

In Figure 9, the entropy for nodes close to the WWTP (node A) was zero, as here no positive signal was detected because of dilution effects. The entropy was also relatively small for the outlet nodes of single sub-catchments (node identifiers starting with S) because these sampling points covered only a few residents. Table 6 shows the stepwise optimization of sampling points with entropy definitions from Section 2.3 and the covered population for each node. The joint entropy reached the maximum value with seven sampling points.

With each added sampling point (s∈S), the joint entropy (JE) and the value of the objective function ($MIMR_{S}$) increase until seven sampling points are reached. The selected candidate nodes as sampling points have the highest information content with minor redundancy for this scenario (detectability of one infected individual out of 2641 noninfected and 10 infected individuals). However, only one sub-catchment was not observed with the defined sampling points, when the entropy reached maximum for this combination of sampling points. One further sampling point must be implemented for this sub-catchment (see candidate node S24 in Figure 9). Subsequently adding the remaining sub-catchment CHE with sampling point (candidate node) S24, the entire population of Hildesheim can be surveilled. Some of the chosen sampling points will provide positive signals with less than ten infected individuals, e.g., for node S23 with 2584 residents, only one infected individual would be sufficient to detect a positive signal.

## 4. Conclusions

This paper develops a pandemic early warning system using samples from a sewer network. This system consists of two sequential parts: a signal detection program based on the mass balance model and an optimal sampling point selection program based on information theory. The signal detection program calculates the theoretical system sensitivity, i.e., the minimum number of infections for which a positive signal can be obtained at a sampling point, standardizing the detectability of sampling points in the WBE, such as the dilution effect. Following this, the actual number of infected persons and the minimum number of sampling points are used to mathematically optimize the location of sampling points in the sewer network, considering the network topology and the settlement structure.

The key findings of this study are as follows:Virus specific parameter values of SARS-CoV-2 from the literature are currently not sufficient for parametrizing our model.Number and locations for the sampling points depends on the expected sensitivity of the system.Increasing the number of sampling points does not necessarily improve the information content.Virus-related uncertainties have an impact on the placement and number of sampling points, but this impact is offset by the expected sensitivity.For the case study of Hildesheim, only 8 sampling points and less than 10 infected individuals per sub-catchment were required to identify potentially infected sub-catchments.

However, some limitations remain in this paper:The probability distribution function is simply based on the assumption that all infected people come from the same sub-catchment. For a better representation, epidemiological data could be used to estimate real infection distributions, as shown in Figure 5b.The flow time used to calculate the system sensitivity simply uses a constant. For further studies, 1D sewer models can be applied to better estimate the flow time and also simulate RNA loss, which is another limitation of the current approach.

However, since our practical detection limits agree with the theoretical calculations, this can be ignored for demonstrating the developed method. However, it is necessary for a more realistic or complex sewer network.

## Figures and Tables

**Figure 1 mps-07-00006-f001:**
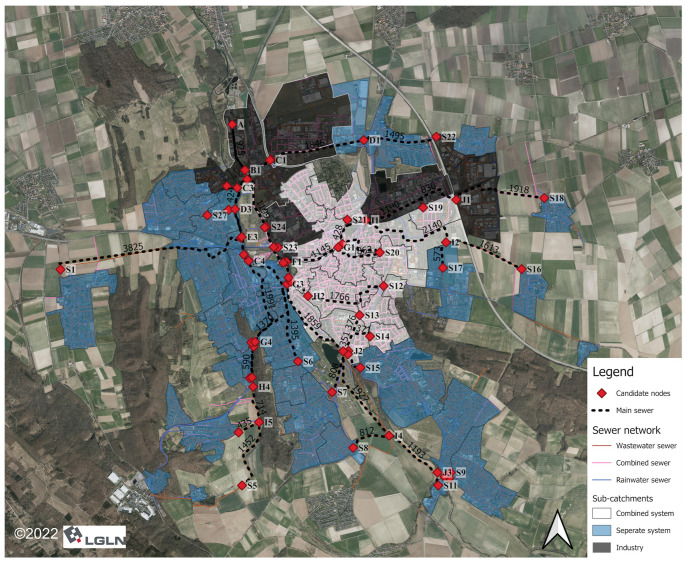
Study area: the main sewer network of Hildesheim with candidate nodes and sub-catchments (source: Landesamt für Geoinformation und Landesvermessung Niedersachsen/LGLN).

**Figure 2 mps-07-00006-f002:**
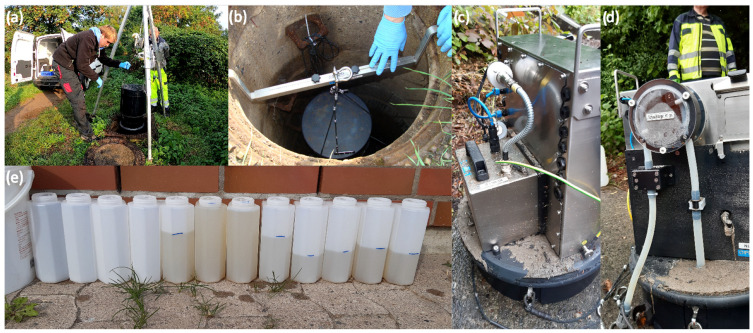
Photos from fieldwork: (**a**,**b**) installation of an autosampler in a manhole; (**c**,**d**) damaged autosampler due to surcharge; (**e**) quality of two-hour composite samples over 24 h from one autosampler.

**Figure 3 mps-07-00006-f003:**
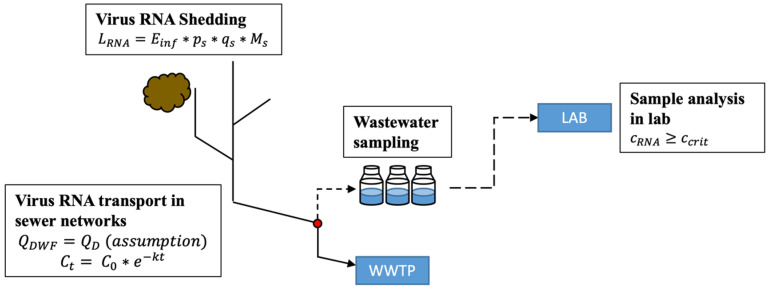
Mass balance model from RNA shedding to sample analysis, where $C_{t}$ and $C_{0}$ are the concentrations of virus RNA in wastewater at time t and time 0, $L_{RNA}$ (copies/day) is the detected RNA load, $C_{RNA}$ (copies/mL) is the detected RNA concentration in the lab, $c_{crit}\;$(copies/mL) is the critical detection limit, $Q_{DWF}$ (mL/day) refers to dry weather flow (only domestic wastewater $Q_{D}$ is considered), $p_{s}$ (-) is the virus RNA shedding probability in stool, $q_{s}$ (mL/(person*day)) is the volume of stool produced per individual and day, $M_{s}$ (copies/mL) is the virus RNA shedding magnitude in stool, k (/day) is the first-order decay value of RNA in wastewater, and t (day) is the flow time of wastewater in the sewer network from RNA input to sampling point.

**Figure 4 mps-07-00006-f004:**
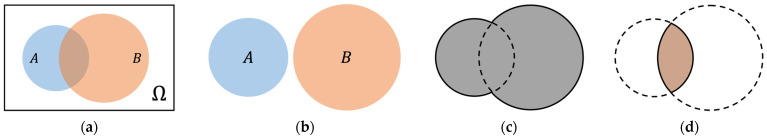
Venn diagrams describing (**a**) the total system, Ω, and two events (A and B), (**b**) individual entropy, (**c**) joint entropy, and (**d**) total correlation [26].

**Figure 5 mps-07-00006-f005:**
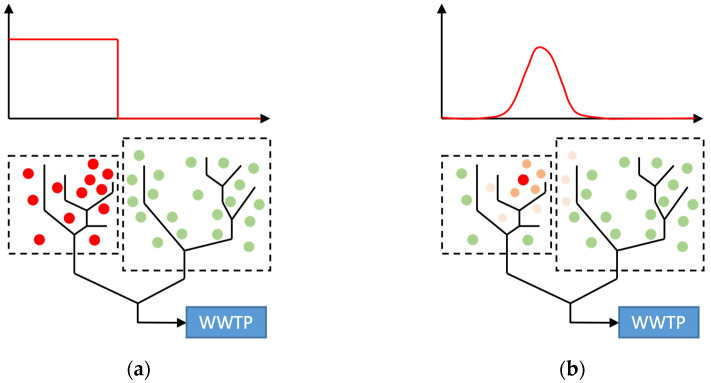
Different definitions of the probability distribution using network topology and settlement structure: (**a**) sub-catchment-dependent probability distribution, (**b**) realistic probability distribution. Dots indicate potentially infected individuals (green dot: the probability that the individual is infected is zero, red dot: the lighter the red, the less likely the individual is to be infected).

**Figure 6 mps-07-00006-f006:**
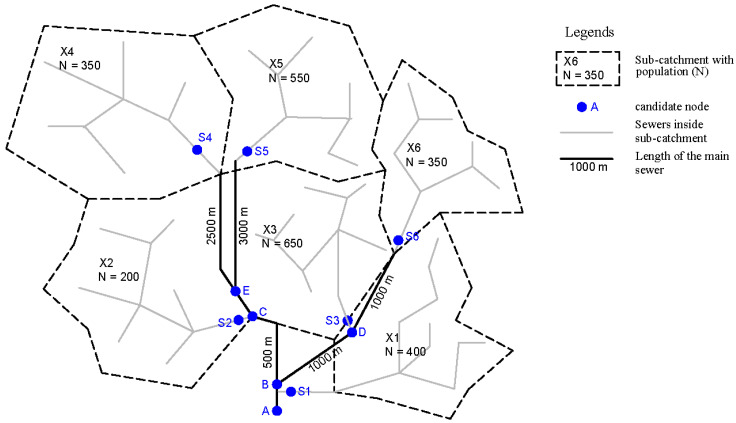
Hypothetical sewer network with 6 sub-catchments and 11 candidate nodes.

**Figure 7 mps-07-00006-f007:**
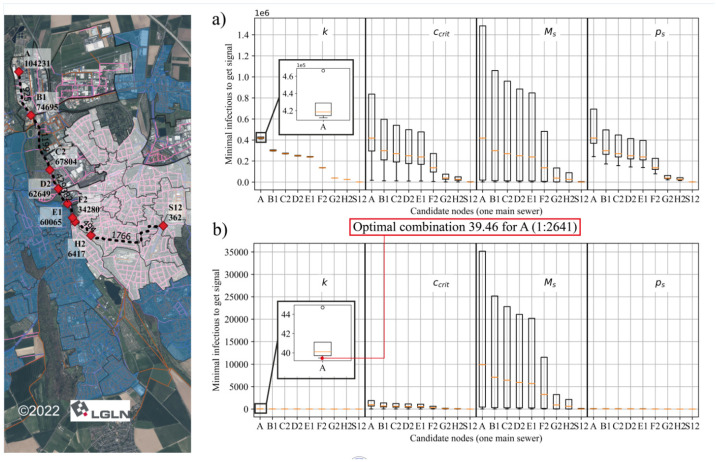
Longitudinal section of candidate nodes (capital letter with number) and covered residents (number below nodes’ identifier). The minimal number of infected individuals to detect a positive signal by changing one parameter and holding (**a**) all other parameters with their median values and (**b**) all other parameters with their optimal values. Virus-dependent parameters: k—Virus RNA decay in wastewater, $c_{crit}$—Critical detection limit, $M_{S}$—Virus RNA shedding magnitude, $p_{S}$—Virus RNA shedding probability.

**Figure 8 mps-07-00006-f008:**
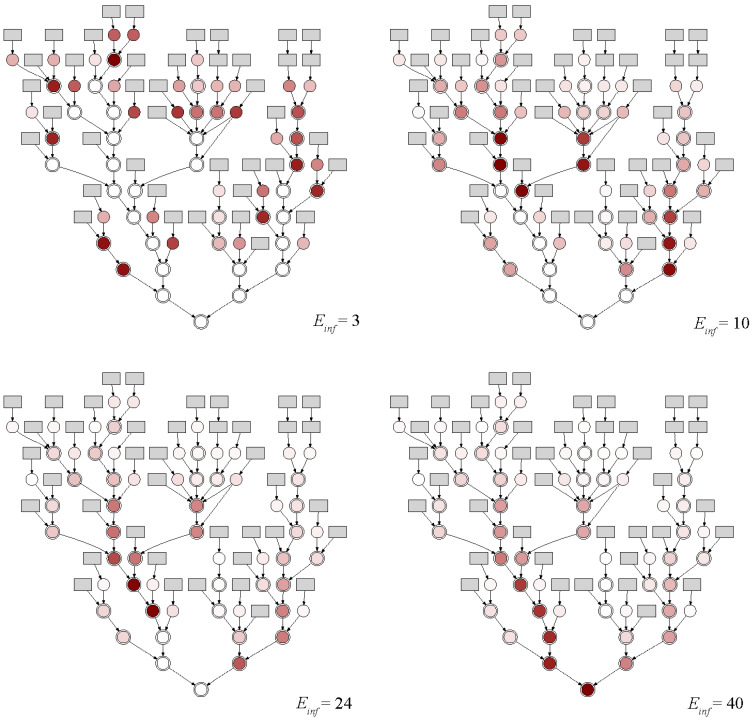
Color maps of entropy based on different $E_{inf}$ within the detection capability 1:2641. The darker the color of the node, the higher the entropy.

**Figure 9 mps-07-00006-f009:**
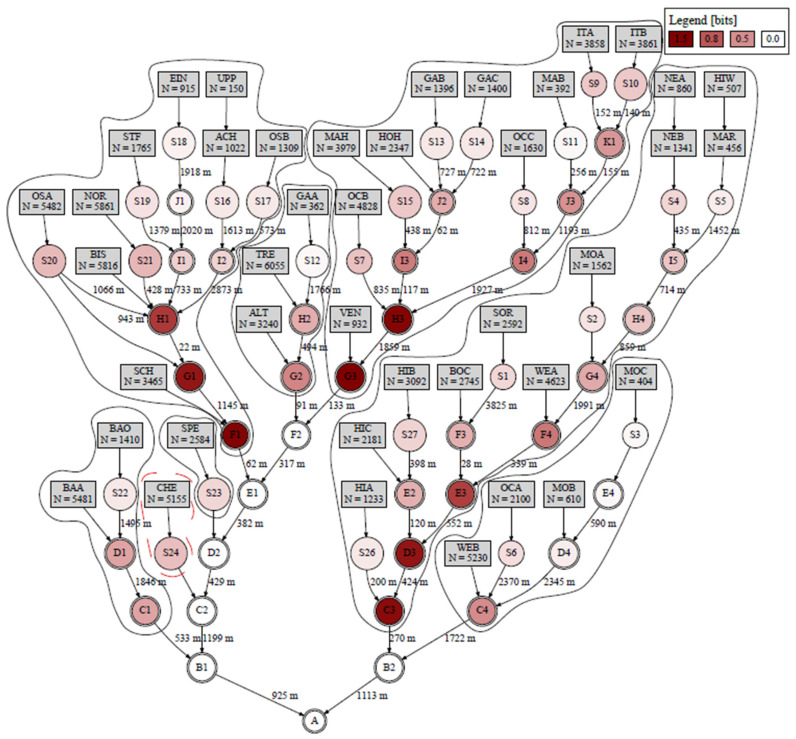
Entropy for each node and optimal sampling point covering the corresponding sub-catchments with eight sampling points (system sensitivity 1:2641, $E_{inf}=$ 10). The numbers of the lines indicate the length of the specific sewer.

**Table 1 mps-07-00006-t001:** Data used, with source.

Data Used	Data Source
Sewer network	Stadtentwässerung Hildesheim (SEHi) (2021)
Population statistics	Stadt Hildesheim (2022), https://www.stadt-hildesheim.de/rathaus-verwaltung/buerger-und-ratsinfo/stadtteile/ (accessed on 1 December 2022)
Land use map	Stadt Hildesheim (2015), https://www.stadt-hildesheim.de/wirtschaft-bauen/stadtplanung-und-stadtentwicklung/stadtentwicklung/flaechennutzungsplan/ (accessed on 1 December 2022)
Digital orthophoto (DOP)	Landesamt für Geoinformation und Landesvermessung Niedersachsen (LGLN) 2022, https://opengeodata.lgln.niedersachsen.de/#dop (accessed on 1 December 2022)
3D building model	LGLN (2022), https://opengeodata.lgln.niedersachsen.de/#lod2 (accessed on 1 December 2022)
ALKIS-Dataset	LGLN (2021), provided by SEHi

**Table 2 mps-07-00006-t002:** Applied values of virus unspecific parameters.

Parameter	Value	Comments	Source
Feces production rate	128 g/(person*day)	Wet mass	[18]
Feces density	1.06 g/mL		[19]
Average water consumption	128 L/(person*day)	The value from 2019	[20]

**Table 3 mps-07-00006-t003:** Statistical information on the virus-dependent parameters from the literature review ^1^.

Factors	Labels	Unit	Min	25%	50%	75%	Max
RNA shedding magnitude	$M_{s}$	$(\log_{10}\mathrm{copies}$/mL)	2.90	4.15	4.70	6.04	**7.10**
RNA shedding probability	$p_{s}$	(%)	10.1	29.0	48.1	54.5	**83.3**
RNA decay in wastewater	k	(/day)	**0.06**	0.09	0.14	0.26	0.67
RNA critical detection limit	$c_{crit}$	(copies/mL)	**3.70**	62.66	88.75	177.28	533.78

^1^ Numbers in bold indicate the best value in each row.

**Table 4 mps-07-00006-t004:** An example of a potential signal matrix using the system sensitivity of 1:2200 based on the hypothetical model with the probability of the infected individual from a specific sub-catchment (+ positive: signal detected on candidate node, − negative: no signal detected).

Source X	Candidate Nodes C											Probability
	A	B	C	D	E	S1	S2	S3	S4	S5	S6	$p(X_{i})$
X1	−	−	−	−	−	+	−	−	−	−	−	0.16
X2	−	+	+	−	−	−	+	−	−	−	−	0.08
X3	−	+	−	+	−	−	−	+	−	−	−	0.26
X4	−	+	+	−	+	−	−	−	+	−	−	0.14
X5	−	+	+	−	+	−	−	−	−	+	−	0.22
X6	−	+	−	+	−	−	−	−	−	−	+	0.14

**Table 5 mps-07-00006-t005:** Relationship between the number of infected individuals in one sub-catchment and entropies ^1^. Numbers in brackets indicate that one more sampling point is needed to cover the catchment. $E_{inf}$ is the number of infected individuals in the sub-catchment, $N_{S}$ is the smallest number of sampling points needed to reach the maximum joint entropy, ${JE}^{MAX}$, $theo.{JE}_{CA}^{MAX}$ is the theoretical maximum joint entropy of the covered areas, $N_{covered}$ is the maximum number of covered populations, and $N_{total}$ is the total number of populations.

$E_{inf}$	$N_{S}$	${JE}^{MAX}$	$\frac{{JE}^{MAX}}{theo.{JE}_{CA}^{MAX}}$	$N_{covered}$	$\frac{N_{covered}}{N_{total}}$
(-)	(-)	(bits)	(%)	(-)	(%)
1	17	1.86	37.4	26,000	24.9
2	20	3.55	71.3	58,046	55.7
3	20	4.43	89.1	80,925	77.6
4	16	4.85	97.4	94,018	90.2
6	15	4.85	97.4	94,018	90.2
8	11	4.95	99.4	99,834	95.8
9	8	4.95	99.4	99,834	95.8
** 10 **	7 **(8)**	** 4.98 **	** 100.0 **	99,076 **(104,231)**	95.1 **(100.0)**
12	6 (7)	4.98	100.0	99,076 (104,231)	95.1 (100.0)
13	5 (6)	4.98	100.0	99,076 (104,231)	95.1 (100.0)
23	4 (5)	4.98	100.0	99,076 (104,231)	95.1 (100.0)
24	3 (4)	4.98	100.0	99,076 (104,231)	95.1 (100.0)
26	3 (4)	4.98	100.0	98,750 (104,231)	94.7 (100.0)
29	2	4.98	100.0	104,231	100.0
40	1	4.98	100.0	104,231	100.0

^1^ Bold and underlined numbers indicate the reached values for the optimal set of sampling points.

**Table 6 mps-07-00006-t006:** Optimization process of sampling points (system sensitivity 1:2641, $E_{inf}=10$).

$N_{S}$	S	JE	TC	$MIMR_{S}$	$N_{covered}$
1	G3	1.51	0.00	1.21	24,623
2	F1	2.88	0.11	2.28	25,785
3	C3	4.02	0.34	3.15	21,192
4	G2	4.40	0.51	3.42	9657
5	C4	4.72	0.71	3.64	8344
6	C1	4.91	0.92	3.74	6891
7	S23	4.98	1.02	3.78	2584
8	S24	4.98	-	-	5155

## Data Availability

Data are contained within the article.

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
