# Peer review of "Optimal Selection of Sampling Points within Sewer Networks for Wastewater-Based Epidemiology Applications"

_mps, 2024, doi:10.3390/mps7010006_

Round 1
Reviewer 1 Report
Comments and Suggestions for Authors
The present study focused I the structure of a tool combines a monitoring model to detect positive signals from regions with infected populations and an optimization program to select optimal sampling points in the sewer system.
The present work present innovation the signal detection program is sensitive and may establish the minimum number of infections for which a positive signal can be obtained at a sampling point, and may put standardization and detectability of sampling points, and the dilution effect.
Also, the work determined actual number of infected persons and the mini-number of sampling points that have to be used by mathematically model able to optimize the location of sampling points in the sewer network, considering the network topology and the settlement structure.
The work present innovation since it treated high number sampling in contrast that was determined in previous work, also the rapid detection and automatized sampling and adoption to a mathematical models with high point samples may confirm the innovative aspect of this work.
1. Page 2: Line 94, Authors describe that the output of wastewater will be the Innerste River, receiving the catchment's water after the WWTP, however they did not precise the pollution degree of this river specially by the biological tools that they consider.
2. Page 3 Line 113, Authors announce the different wastewater matrices (Figure 2e) can affect the detection of RNA by this remark can this be a week point in their model that they introduce, please explain. Also, I suppose that the RNA and environment interaction may affect the system and the model proposed we suppose that the sensibility of RNA in stressed conditions and may also affect the system sensibility.
Also, some interesting references that deserve to be cited in Bibliography were be missed such as :
Gkatzioura, A.; Zafeirakou, A. Optimal Selection of Sampling Points for Detecting SARS-CoV-2 RNA in Sewer System Using NSGA-II Algorithm. Water 2023, 15, 4076. https://doi.org/10.3390/w15234076
Reviewer 2 Report
Comments and Suggestions for Authors
The manuscript, submitted by Yao and colleagues for publication to the journal Methods and Protocols, provided a tool for combining a monitoring model and an optimization 21 model in order to facilitate the identification of suitable sampling points in sewer surveillance. They validated this method on real data from Hildesheim, Germany focusing on SARS-CoV-2 surveillance. The methods are very interesting, while the main text is well structured and exhaustive in every part. It can be recommended for publication, upon addressing some minor revisions.
The following points have been listed to evaluate a possible optimization, about the text and the images, for the article to which the reference is made.
1) In the abstract section at LINE 23: “utilizes the parameters from previous studies”, since generally in the abstract section the references are not mentioned, I would sugget to modify this sentence.
2) FIGURE 1: the figure reports the different areas of sub-catchments, but the difference between the colours used for “combined system” and “industry” is not well evident.
3) LINE 138: “ultimately” could be replaced with “in the second step” to help the reader to not lose the thread of the conversation.
4) FIGURE 3: In the caption, the different variables described could be put in bold or in a list to be easier to read.
5) LINE 205: on the base of the definition of TC, the variables of the difference in the equation 4 shouldn’t be inverted? Furthermore, the value of the difference should be a negative number, isn’t it?
6) LINE 283: The error is referred to the equation 4?
7) LINE 290: Instead of “has to be” the correct meaning of the sentence could be given with “has been”?
8) LINE 340: Is indicated the section “4.2”, but maybe the correct section is “3.2”?
9) LINE 341: Instead of “independent” I would suggest to change in “regardless of”?
Comments on the Quality of English LanguageThere are some typos. I would suggest a extensive editing of English language.
Author Response
We would like to thank the anonymous reviewer for their valuable comments that we took carefully into account. The revised parts of the manuscript are all highlighted in yellow.
- In the abstract section at LINE 23: “utilizes the parameters from previous studies”, since generally in the abstract section the references are not mentioned, I would sugget to modify this sentence.
The text in lines 22-24 is modified.
- FIGURE 1: the figure reports the different areas of sub-catchments, but the difference between the colours used for “combined system” and “industry” is not well evident.
Corrected.
- LINE 138: “ultimately” could be replaced with “in the second step” to help the reader to not lose the thread of the conversation.
Corrected.
- FIGURE 3: In the caption, the different variables described could be put in bold or in a list to be easier to read.
Corrected. Also, for Figure 7 and Table 5.
- LINE 205: on the base of the definition of TC, the variables of the difference in the equation 4 shouldn’t be inverted? Furthermore, the value of the difference should be a negative number, isn’t it?
Equation 4 is in accordance with the paper (https://agupubs.onlinelibrary.wiley.com/doi/full/10.1029/2011WR011251).
- LINE 283: The error is referred to the equation 4?
Same as 5.
- LINE 290: Instead of “has to be” the correct meaning of the sentence could be given with “has been”?
Corrected.
- LINE 340: Is indicated the section “4.2”, but maybe the correct section is “3.2”?
Corrected.
- LINE 341: Instead of “independent” I would suggest to change in “regardless of”?
Corrected.

Reviewer 3 Report
Comments and Suggestions for Authors
The manuscript entitled "Optimal Selection of Sampling Points Within Sewer Networks for Wastewater-Based Epidemiology Applications" delivers a comprehensive and innovative exploration of pandemic early warning systems using sewer network samples. The integration of a signal detection program based on mass balance modeling and an optimal sampling point selection program grounded in information theory constitutes a novel and valuable contribution to the field of wastewater-based epidemiology.
Nevertheless, some points require improvement.
The abstract should be clear and include context, objectives, and methodology. Explain "Sewer Net-2 Works" and the monitoring model's virus detectability parameters. Keep sentences concise while retaining important information. Emphasize validating the tool with SARS-CoV-2 and its adaptability for monitoring various pathogens. Expand keywords to include "pathogen surveillance" and specify.
Authors should provide a more concise overview of the challenges associated with WBE, emphasize specific hurdles addressed in their study, define "Sewer Net-2 Works" more clearly, make the transition between challenges and mass balance model smoother, and provide a more compelling rationale for selecting sampling points within sewer networks. The paragraph about the tool's structure and comparison with previous methods is informative but could be streamlined. Lastly, authors might consider incorporating a brief preview of key findings or contributions of the study. In the materials and methods section, the rationale for using Equation (5) for probability distribution could be clarified. The results section provides a robust analysis of experimental results, effectively connecting the determination of system sensitivity with the optimization of sampling points. Improvements could involve streamlining some technical details to enhance readability. The figures presented are both scientifically and graphically accurate.
In the Conclusions section, would be beneficial to provide a brief recap of the key findings and contributions before delving into the limitations. Each limitation could be discussed separately, followed by potential avenues for future research.
Despite these considerations, the manuscript presents a well-structured and methodically executed study that significantly advances our understanding of optimizing sampling points in sewer networks for effective early detection of potential outbreaks. The English language quality is commendable, and the manuscript effectively communicates the research findings and methodologies in a formal and academic style.
